# SONAR: A Physics-constrained Neural Representation for X-ray Dark-field CT

**Daniel Frey**[*†1,2] (iD)                                    DANIEL.FREY@TUM.DE

**Theresa Hiu**[*1,2]                                         THERESA.HIU@TUM.DE

**Julian McGinnis**[3,4] (iD)                                JULIAN.MCGINNIS@TUM.DE

**Tina Dorosti**[1,2,5] (iD)                                  TINA.DOROSTI@TUM.DE

**Johannes Thalhammer**[1,2,5,6] (iD)            JOHANNES.THALHAMMER@TUM.DE

**Sebastian Peterhansl**[1,2]                       SEBASTIAN.PETERHANSL@TUM.DE

**Zijin Huang**[1,2]                                           ZJ.HUANG@TUM.DE

**Franz Pfeiffer**[1,2,5,6] (iD)                           FRANZ.PFEIFFER@TUM.DE

**Daniel Rueckert**[3,4,7] (iD)                        DANIEL.RUECKERT@TUM.DE

**Florian Schaff**[1,2] (iD)                               FLORIAN.SCHAFF@TUM.DE

[1] *Chair of Biomedical Physics, TUM School of Natural Sciences, TUM*

[2] *Munich Institute of Biomedical Engineering, TUM*

[3] *Chair of AI in Healthcare and Medicine, TUM University Hospital, TUM*

[4] *Munich Center for Machine Learning (MCML), TUM*

[5] *Institute for Diagnostic and Interventional Radiography, TUM University Hospital, TUM*

[6] *TUM Institute for Advanced Study, TUM*

[7] *Department of Computing, Imperial College London*

## Abstract

Dark-field computed tomography (DFCT) enables functional lung imaging with small-angle X-ray scattering, but reconstructions are often degraded by streak artifacts. We propose SONAR (Shot-Optimized Neural Adaptive Representation), a projection-based implicit neural representation (INR) that jointly models transmission, phase shift, and dark-field signals across neighboring shots using a physics-based Talbot–Lau interferometer forward model. By leveraging adaptive per-projection optimization, SONAR effectively enables stabilized phase retrieval and suppresses streak artifacts for improved DFCT image quality on a grating-based human-scale prototype.

**Keywords:** Computed tomography, dark-field imaging, implicit neural representations

## 1. Introduction

Human-scale dark-field computed tomography (DFCT) has recently been realized using a grating-based gantry prototype (Viermetz et al., 2022). This technique shows strong clinical potential, however, its continuous rotation poses a crucial challenge for phase retrieval, which can result in pronounced streak artifacts (Viermetz et al., 2023; Schmid et al., 2023; Haeusele et al., 2023). Convolutional neural networks (CNNs) have shown promise for DFCT image enhancement, but training data scarcity and the prospective domain gap limit

---

[*] Contributed equally

[†] Corresponding author

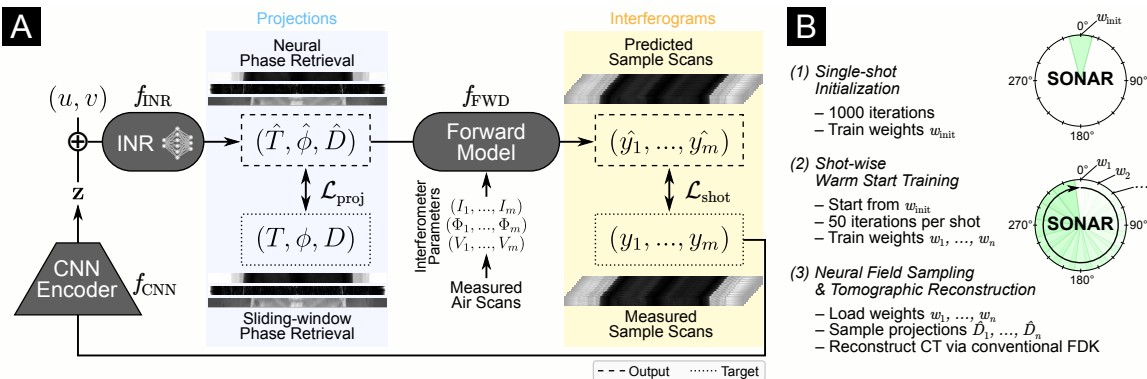

Figure 1: SONAR framework. (A) Neural field representation modeling transmission $T$, phase shift $\phi$, and dark-field $D$ projections. (B) Training schedule across all shots acquired with a Talbot–Lau grating interferometer.

the applicability of supervised approaches (Kumschier et al., 2024). In contrast, implicit neural representations (INRs) are trained on a per-instance basis to learn a continuous mapping of spatial image coordinates to intensity values, sinusoidal representation networks (SIRENs) being particularly popular for improved high-frequency fidelity (Sitzmann et al., 2020). In this work, we introduce a physics-constrained neural field jointly representing grating-based X-ray projection contrasts for streak-reduced DFCT reconstruction.

## 2. Methods

A conceptual sketch of SONAR is provided in Fig. 1. We modeled transmission $T$, phase shift $\phi$, and dark-field $D$ projections using a tri-head SIREN

$$f_{\text{INR}} : (u, v, \mathbf{z}) \mapsto (T, \phi, D), \tag{1}$$

where $u$ and $v$ are detector coordinates and $\mathbf{z}$ is a latent code conditioned on measured interferograms, encoded via a lightweight CNN for spatial image coherence. For each projection, the corresponding stack of interferograms was encoded via $f_{\text{CNN}}$, resampled to match coordinate dimensions and concatenated into a combined input for the INR. Given $I$, $\Phi$, and $V$, derived from air scans and denoting the mean intensity, the phase of the fringe pattern, and the fringe amplitude (visibility), we synthesized

$$f_{\text{FWD}} : (T, \phi, D) \mapsto y = TI + TIDV \cos(\Phi + \phi), \tag{2}$$

approximating interferometer physics with a truncated Fourier series (Haeusele et al., 2024). We optimized the INR directly in measurement space using $m = 12$ stacked neighboring shots, which can be considered equivalent to phase stepping across a full interferometer period (Viermetz et al., 2022). We trained the INR per projection using a combined loss, where $\mathcal{L}_{\text{shot}}$ enforces consistency with measured interferograms and $\mathcal{L}_{\text{proj}}$ weakly supervises $(T, \phi, D)$ using classical sliding-window phase retrieval (SPR) (Haeusele et al., 2023). Leveraging shared features and angular consistency, we adopted a sequential warm start, initializing each projection from the previous one with few refinement steps.

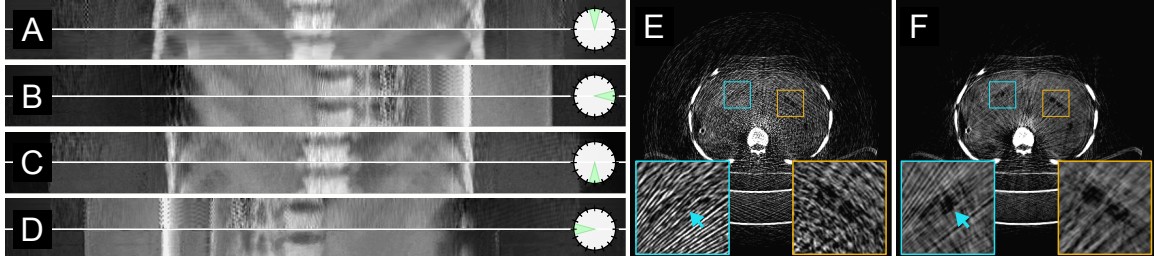

Figure 2: DFCT of porcine lungs inside of a thorax phantom. (A–D) Dark-field projections at 0°, 90°, 180°, and 270°, comparing classical SPR (top) and SONAR (ours, bottom). (E) FDK reconstruction from SPR. (F) FDK reconstruction from SONAR indicating recovered structural details at reduced noise and artifact levels.

## 3. Results and Discussion

SONAR is evaluated on a human-scale grating-based DFCT scan comprising ex vivo porcine lungs ventilated and within a thorax phantom as a clinical surrogate. The neural dark-field projections and corresponding Feldkamp–Davis–Kress (FDK) reconstruction are shown in Fig. 2, compared against a conventional sliding-window baseline. Across projections, our method substantially reduces noise and artifacts, particularly in low-contrast and edge regions where the baseline degrades due to cross-talk artifacts (Haeusele et al., 2024). This improvement is consistently reflected in the reconstructed DFCT volume, where SONAR yields sharper and more coherent structural features while strongly suppressing streaks. Because our model is trained in the projection domain and restricted to physically plausible interferometric solutions, the risk of hallucinated structures is substantially reduced. Overall, our results suggest that the low-rank structure of dark-field signals aligns well with the spectral bias of neural fields (Rahaman et al., 2019), enabling recovery of structurally consistent details from limited phase information. Our work demonstrates a physics-constrained neural representation for DFCT that improves reconstruction quality in a reliable, physically consistent, and self-supervised manner without requiring training data. Rather than competing with classical processing, SONAR integrates conventional phase retrieval for continuous-acquisition scans. Future work will focus on assessing hyperparameter robustness across different sample scans and extending the framework to limited- or sparse-view angular representations ensuring rotational consistency and computational efficiency.

## Acknowledgments

Financial support through the European Research Council (ERC Smart Detectors for Dark-field X-ray Imaging, SyG 101167328), and the Free State of Bavaria under the Excellence Strategy of the Federal Government and the States, as well as by the Technical University of Munich – Institute for Advanced Study.

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
