# OpenReview forum: "SONAR: A Physics-constrained Neural Representation for X-ray Dark-field CT"
_MIDL.io/2026/Short_Papers — MIDL 2026 - Short Papers Poster_

### Official Review · Reviewer_avuo · 2026-05-04
**Physics-constrained neural representation for dark-field CT reconstruction**

**Rating:** 5
**Confidence:** 5

**Review:**

Overall, this paper presents a well-motivated and technically solid approach that integrates physics-based modeling with implicit neural representations for DFCT reconstruction. The formulation is elegant and aligns naturally with the underlying imaging physics.

Strengths:
- Strong integration of physics constraints via the Talbot–Lau forward model
- Use of INRs is well-suited for continuous projection modeling
- Self-supervised formulation avoids reliance on large training datasets
- Shot-wise optimization with warm start is practical and effective
- Clear qualitative improvements in artifact suppression and structural coherence

Weaknesses:
- The experimental evaluation is limited, primarily relying on a single dataset and qualitative comparisons. Additional quantitative analysis and broader validation would strengthen the conclusions

Overall, the method is conceptually sound and promising, particularly for challenging DFCT reconstruction settings.

**Summary:**

This paper proposes SONAR, a physics-constrained implicit neural representation (INR) for X-ray dark-field CT reconstruction. The method jointly models transmission, phase shift, and dark-field signals using a SIREN-based neural field, optimized directly in projection space under a Talbot–Lau interferometer forward model. A shot-wise training strategy with warm-start initialization enables stable phase retrieval across projections. Experiments on a human-scale DFCT prototype show improved noise suppression and reduced streak artifacts compared to classical sliding-window phase retrieval, leading to more coherent reconstructions. The approach is self-supervised and leverages physical constraints to reduce hallucination risk.

**Strengths:**

The paper introduces a well-designed physics-constrained INR framework tailored to dark-field CT, effectively combining implicit neural representations with a principled forward model. The self-supervised formulation is particularly appealing in this domain where training data is scarce. The shot-wise optimization strategy is practical and leverages inter-projection consistency. Qualitative results demonstrate clear improvements in noise reduction and artifact suppression, and the approach reduces the risk of hallucinations by enforcing physical consistency.

**Weaknesses:**

The experimental evaluation is somewhat limited, as it is primarily based on a single dataset and relies heavily on qualitative comparisons. While the improvements are visually convincing, additional quantitative metrics and validation across different acquisition settings or samples would further strengthen the evidence and generality of the approach.

**Justification Of Rating:**

The paper presents a technically sound and well-motivated method with clear qualitative improvements and strong alignment with imaging physics. While the evaluation is somewhat limited, the contribution is meaningful and relevant, making it suitable for acceptance.

---

### Decision · Program_Chairs · 2026-05-08

Accept (Poster)